# Utilization of Spectral Indices for High-Throughput Phenotyping

**DOI:** 10.3390/plants11131712

**Published:** 2022-06-28

**Authors:** Rupesh Tayade, Jungbeom Yoon, Liny Lay, Abdul Latif Khan, Youngnam Yoon, Yoonha Kim

**Affiliations:** 1Department of Applied Biosciences, Kyungpook National University, Daegu 41566, Korea; rupesh.tayade@gmail.com (R.T.); layliny22@gmail.com (L.L.); 2Horticultural and Herbal Crop Environment Division, National Institute of Horticultural and Herbal Science, Rural Development Administration, Wanju 55365, Korea; beomi7944@korea.kr; 3Department of Engineering Technology, University of Houston, Texas, TX 77204, USA; latifkust@gmail.com; 4Crop Production Technology Research Division, National Institute of Crop Science, Rural Development Administration, Miryang 50424, Korea

**Keywords:** hyperspectral image, vegetation indices, high-throughput phenotyping, remote sensing, unmanned aerial vehicles

## Abstract

The conventional plant breeding evaluation of large sets of plant phenotypes with precision and speed is very challenging. Thus, consistent, automated, multifaceted, and high-throughput phenotyping (HTP) technologies are becoming increasingly significant as tools to aid conventional breeding programs to develop genetically improved crops. With rapid technological advancement, various vegetation indices (VIs) have been developed. These VI-based imaging approaches, linked with artificial intelligence and a variety of remote sensing applications, provide high-throughput evaluations, particularly in the field of precision agriculture. VIs can be used to analyze and predict different quantitative and qualitative aspects of vegetation. Here, we provide an overview of the various VIs used in agricultural research, focusing on those that are often employed for crop or vegetation evaluation, because that has a linear relationship to crop output, which is frequently utilized in crop chlorophyll, health, moisture, and production predictions. In addition, the following aspects are here described: the importance of VIs in crop research and precision agriculture, their utilization in HTP, recent photogrammetry technology, mapping, and geographic information system software integrated with unmanned aerial vehicles and its key features. Finally, we discuss the challenges and future perspectives of HTP technologies and propose approaches for the development of new tools to assess plants’ agronomic traits and data-driven HTP resolutions for precision breeding.

## 1. Introduction

By 2050, the human population is projected to reach 9.7 billion, and the increasing food demand for the rising population would require a 25% to 70% increase in crop production compared to its current levels [1]. Achieving this target, however, poses a challenge to plant breeders, because food production will have to grow at a rate of 2.4% each year, but the actual growth rate is only 1.3%, with yields stagnating in up to 40% of cereal fields [2]. To increase crop production while reducing its impact on the environment, sustainable intensification in agriculture needs to be developed by adopting modern techniques [3]. One of the easiest ways to maximize the efficiency of major food and feed crops worldwide is to boost plant breeding and propagation.

In recent years, high-throughput phenotyping (HTP) has attracted considerable attention, leading to the creation of many new protocols for documenting different plant traits of particular interest [4]. The crop improvement research field, however, must adapt to address the growing threats posed by climate change [5]. The effects of climate change vary depending on geographical areas, demanding targeted strategies for the world’s numerous agroecological regions. Due to inadequate access to capital and the adoption of less productive farming methods, farmers from low-income countries are highly vulnerable to climate change [6]. As the growing crop production will increase the environmental footprint of agriculture, a sustainable intensification of agricultural practices will be required. Agricultural practices including modern plant breeding, crop cultivation, crop management [3], and HTP methods are ideally suited for improving crop productivity, as well as developing climate-resilient crops. Due to constant environmental changes and their impact on plant growth, it is necessary to gather data from more plant samples to identify common structural and physiological features. For this reason, HTP aims to characterize a large number of populations accurately [7]. Together with HTP, modern plant breeding or new genomics tools, such as genomic selection technology, have been considered the frontiers of crop breeding programs [8]. The current progress in sensors, aeronautics, and computing systems are paving the way for the development of effective field-based phenotyping platforms [9].

Most common HTP platforms consist of the utilization of unmanned vehicles containing various image sensors which are: (i) red-green-blue (RGB) imaging, (ii) infrared (IR) imaging, (iii) hyperspectral imaging, (iv) thermal imaging, (v) fluorescence imaging, (vi) light detection and ranging (LiDAR), and (vii) satellite imaging [10] (Figure 1). Among them, RGB and hyperspectral imaging tools are widely used to evaluate the quantitative and qualitative attributes of plants. In most cases, raw image data need to be converted to other indexes, because plants show different reflectance values at specific wavelengths depending on their growth stages and conditions (Kim et al., 2021). Typically, vegetation indices (VIs) are broadly used to measure plant health and growth conditions (soil property, water or moisture, and nutrient content).

VIs represents the statistical transformation of the initial spectral reflectance, and they are obtained using two or more wavebands to measure and interpret the incidence and condition of vegetation, such as the biomass and canopy attributes [11,12,13,14]. These indices allow reliable spatial and temporal intercomparisons of terrestrial photosynthetic activity and canopy structural variations [14]. The VIs obtained from remote sensing-based canopies are simple and effective algorithms for the quantitative and qualitative evaluation of vegetation cover, robustness, and growth dynamics. As they are simple transformations of spectral bands, they are computed directly without any bias or assumptions regarding land cover class, soil type, or climatic conditions. VIs assist researchers in monitoring seasonal, interannual, and long-term structural variations, as well as the phenological and biophysical parameters of vegetation cover [14]. As previously mentioned, VIs can predict plant growth conditions using numerous spectral reflectance bands, which can easily be acquired by spectral cameras, therefore they are particularly adequate in HTP. However, there is limited information about the use of VIs in the agricultural sector for multiple traits (physiological, biotic and abiotic), replicated multi environment and large populations evaluation. Thus, it is of great interest to understand different VI and HTP applications for crop improvement. In this study, we provide an overview of VIs and HTP and summarize the utilization of VIs for HTP in the agricultural research on the basis of an intensive literature survey.

## 2. Types of Vegetative Indices in Crop Research

In general, most substances including plants have three interactions/responses to electromagnetic energy (for example light) that is reflected, transmitted, or absorbed. In other words, each material or vegetation has its spectral signature/reflectance which can be measured in three key wavelength bands (Figure 1). In the case of plants, the unique interaction with solar radiation distinguishes them from other natural materials. Plants use photosynthesis to generate energy and, as a result, absorb a lot more blue and red light. On the other hand, they strongly reflect green and near-infrared (NIR) light [10]. Plants consist of various materials—such as water, nutrients, and pigments—therefore, they present variations across the spectrum, which provide important information related to water, nutrient, and pigment contents [10,16]. The spectrum variation is often described as VIs for the prediction of plant growth conditions [11].

Over the years VI research has evolved, and numerous new VIs have been developed to understand the aerial imagery included in the literature [17,18,19,20]. The chronological information details of most of the vegetation indices developed and reported by researchers are presented in (Table 1). Among them, 25 are broadly used in plant research because they are very effective in determining various features, such as leaf greenness, light use efficiency, leaf pigment, and water content [10,20]. Most VIs are calculated by two or more reflectance wavelengths, which are normally involved in the concentration of photosynthetic pigments [10]. However, among these VIs, very few have been critically compared or tested. Therefore, here we describe several selected indices that are most often used for crop or vegetation evaluation, and that have the linear link with crop yield, pigments, health, and canopy water content, and are also often used in agricultural production estimation.

### 2.1. Determination of Leaf Greenness

The VIs associated with leaf greenness determine photosynthetic pigments, which are present in green vegetation. They record chlorophyll and utilize the red and NIR wavelength spectrum to assess green vegetation. The following two VIs are widely used by researchers as leaf greenness determinants.

**i.** Normalized difference vegetation index (NDVI): Typically a simple and suitable indicator, first proposed by Rouse Jr. et al. [97], this index calculates and differentiates vegetation from non-vegetation zones based on reflectance in the red and NIR wavelengths. NDVI is sensitive to chlorophyll content. The values of this index are in the range (−1.0–1.0). The common range for healthy green vegetation is 0.2 to 0.9. The following Equation (1) is used for its calculation:(1)$$\mathrm{NDVI}=\frac{R_{NIR}-R_{RED}}{R_{NIR}+R_{RED}}$$
where *R*_*NIR*_ and *R*_*RED*_ represent *NIR* wavebands and spectral reflectance in the red respectively.

**ii.** Modified chlorophyll absorption in reflectance index (MCARI): MCARI is an alternative index that was originally defined by [98], and was mainly developed to study variations in chlorophyll content. However, researchers consider MCARI as very sensitive to low chlorophyll content, as it is influenced by non-photosynthetic materials/pigments, background noise, and reflectance. The value of this index ranges from −1 to 1. The common range for healthy green vegetation is 0.2 to 0.7. The following Equation (2) is used to define this index:(2)$$\mathrm{MCARI}=[\left(R_{700}-R_{670}\right)-0.2\;\times \;\left(R_{700}-R_{550}\right)]\;\times \;\left(\frac{R_{700}}{R_{670}}\right)$$
where *R* denotes the reflectance for chlorophyll absorption at the 670 nm waveband relative to the reflectance at the 550 nm and 700 nm wavebands.

### 2.2. Determination of Light Use Efficiency

The VIs used to determine light use efficiency are indicators of how well vegetation can absorb incident light to perform photosynthesis. These VIs are useful in precision agriculture, as they assist in the estimation of growth and productivity. In addition, they utilize visible spectrum reflectance measurements to assess the vegetation’s total light consumption efficiency by taking into account interactions between different pigments. A couple of well-established light use efficiency VIs are mentioned below.

**i.** Photochemical reflectance index (PRI): PRI is a reflectance assessment of green vegetation that is sensitive to variations in carotenoid pigments, particularly xanthophyll. Changes in carotenoid pigments are indicative of per unit energy absorbed during photosynthesis as light use efficiency, or of the rate of carbon dioxide absorbed by green vegetation. The range for a PRI is −1 to 1, where healthy vegetation generally often ranges between −0.2 and 0.2. PRI was originally known as the “physiological reflectance index” [99,100,101] and the equation below is used for its calculation:(3)$$\mathrm{PRI}=\frac{R_{531}-R_{570}}{R_{531}+R_{570}}$$
where *R* denotes the reflectance at the waveband of concern, particularly light absorption for xanthophyll at the 531 nm waveband, and to minimize the effects of chloroplast mobility, a reference waveband was employed at 570 nm.

**ii.** Red-green ratio index (RGRI): This index uses the broad wavelengths called red band (600–699 nm), and green band (500–599 nm), and is a reflectance measurement that compares anthocyanin-induced leaf redness to chlorophyll. The red-green ratio has been used to forecast the development of vegetation in canopies, and also to determine leaf production and stress and, in some instances, flowering. The range of an RGRI is from 0.1 to >8, where healthy green vegetation usually falls between values of 0.7 to 3. This index is defined by the equation below proposed by [78]:(4)$$RGRI=R_{\mathrm{RED}}:R_{\mathrm{GREEN}}$$
where *R* is the reflectance, *RED* refers to a broad range red band at 600–699 nm, and *GREEN* to a broad range green band at 500–599 nm.

### 2.3. Determination of Leaf Pigment

Leaf pigment VIs are used to determine pigments (anthocyanin and carotenoid), which are present in higher amounts in stressed vegetation, and they do not record chlorophyll. The following two VIs are widely used as leaf pigment determinants.

**i.** Anthocyanin reflectance index 1 (AR1): This index is based on reflectance measurements in the visible spectrum and quantifies stressed vegetation. Typically, alterations of, or increases in ARI1 suggest canopy modification caused by either new growth or death. The values of this index ranged from 0 to 0.2; however, it can vary with the plant species from −∞ to ∞. The estimate of anthocyanin accumulation is calculated by the equation below described in [21]:(5)$$\mathrm{ARI}1=\frac{1}{\;\;R_{550}}-\frac{1}{R_{700}}$$
where *R* is the reciprocal reflectance at the waveband of concern, 550 nm and 700 nm.

**ii.** Carotenoid reflectance index 1 (CRI1): this index is considered as a reflectance measure sensitive to the carotenoid pigments found in plant leaves and is mostly applicable in the measurement of stressed vegetation. Typically, stressed or weak vegetation presents a greater accumulation of carotenoids. The values of this index range from 0 to >15. The common range for green vegetation is between values of 1 and 12. CRI1 is calculated based on the well-defined equation mentioned below [24]:(6)$$\mathrm{CRI}1\;=\;\frac{1}{R_{510}}-\frac{1}{R_{550}}\;$$
where *R* is the reciprocal reflectance at the waveband of concern; at 510 nm both carotenoids and chlorophyll affect reflectance, and at 550 nm, only chlorophyll affects reflectance.

### 2.4. Determination of Canopy Water Content

The VIs that determines canopy water content use reflectance measurements in the near-infrared and shortwave infrared ranges. These VIs provide an indication of how much water is contained within the leafy canopy. The presence of a high water content in vegetation suggests that plants are healthier and more likely to grow faster and tolerate extreme conditions (high temperature, fires, and drought). The following VIs are widely applicable for the measurement of canopy water content.

**i.** Moisture stress index (MSI): This index is linearly correlated to the relative water content in leaves. Unlike in other water VIs, the relationship between values of the MSI is reversed; higher values indicate a higher water stress and a lower water content. The values of this index range from 0 to >3. The common range for green vegetation is 0.4 to 2. The following equation is used for the calculation of this index [60,61]:(7)$$\mathrm{MSI}=\left(\frac{R_{1599}}{R_{819}}\right)$$
where *R* is the reflectance at the waveband of concern; water absorbs shortwave infrared light at wavebands around 1599 nm. Absorption at 819 nm is used as a reference since it does not get affected by changes in water content.

**ii.** Normalized multiband drought index (NMDI): This index was initially proposed for monitoring soil and vegetation moisture from space, as was described in [71]. Index values range from 0.7 to 1 for dry soil, 0.6 to 0.7 for soil with intermediate moisture, and less than 0.6 for wet soil. It was developed based on three different wavelengths (NIR, centered around 860 nm, and two others in the shortwave infrared, centered around 1640 nm and 2130 nm), and is calculated as follows:(8)$$\mathrm{NMDI}=\frac{R_{860}-(R_{1640}-\rho_{2130})}{R_{860}+(R_{1640\;}+\rho_{2130})}$$
where *R* is the reflectance at the waveband of concern, particularly reflectance observed by a satellite sensor at the 860 nm, 1640 nm, and 2130 nm bands, respectively. It uses the difference (slope) between two absorption bands (1640 nm and 2130 nm), as the soil and vegetation water-sensitive band.

Apart from the above-mentioned widely used VIs, with recent technological advancement, several additional VIs have also been developed. Examples include the ratio spectral index and normalized difference spectral index, which are used for recognizing vegetation stress caused by hydrocarbon pollution [102], the enhanced vegetation index (EVI) [14,103], the perpendicular vegetation index, and the atmospherically resistant vegetation index (ARVI) [104]. Furthermore, these VIs improve sensitivity in high biomass environments by reducing soil background (brightness, color) and other atmospheric impacts (cloud, leaf canopy shadow, and cloud shadow). However, different VIs have their own distinct set of spectra that reveal information regarding specific plant properties. Depending on individual research objectives or interests, these VIs can be integrated into agriculture for both quantitative and qualitative evaluation of vegetation to obtain precise information about crops. However, most of these indices have indeed been evaluated only for a few species, so it is unclear whether they will be applicable across a wide range of plant species with different leaf characteristics and phenotypic, or morphological, expression. Thus, further comparative and validation studies of the numerous VIs developed for crop assessment are needed.

## 3. Utilization of Vegetation Indices in Crop Research

### 3.1. Importance in Breeding Programs

The continuous advancement of breeding techniques allows an increase in the pace of genetic enhancement [105]. The selection of desired plants based on phenotypes has been performed by farmers and plant breeders long before DNA and molecular markers were discovered. Crop breeding is a game of numbers: the more crosses and conditions used for selection, the higher the chance of superior varieties to be found. Plant breeders must be able to determine the best progeny easily and specifically, by phenotyping vast numbers of lines. There is a need to improve breeding productivity to satisfy potential future needs. Advances in high-throughput genotyping have produced fast and inexpensive genomic knowledge, and this technology has paved the way for the creation of thousands of recombinant inbred lines for phenotyping broad mapping populations and diversity panels [106]. As molecular breeding strategies (such as marker-assisted recurrent selection and genomic selection) concentrate mostly on genotypic information-based choices, phenotypic information is still needed [107]. Similarly, to classify promising events in transgenic experiments, phenotyping is also needed [108,109]. To capitalize on advances in traditional, molecular, and transgenic breeding, and ensure the genetic enhancement of crops, successful phenotyping is likely to be necessary. The demand for efficient phenotyping methods has been introduced in numerous domains. In many cases, phenotypes are robust predictors of important biological traits, such as disease and mortality [110]. Molecular biologists and breeders believe that advanced molecular techniques can only be useful in breeding if the collection of quantitative traits is based on reliable phenotyping techniques [111]. High-performance phenotyping techniques can transform the plant breeding process by accelerating the generation advancing process [8].

### 3.2. Importance in Precision Agriculture

Overall, yield and plant performance are linked to traditional phenotypes—such as seedling vigor, days to flowering time, and terminal plant height—and a considerable amount of research tried to measure these phenotypes by HTP [8,9,16,112]. Therefore, more advancements in phenotyping techniques are needed to enhance breeding programs, accelerate genetic gains, and simplify the tracking of plants’ health status to minimize qualitative and quantitative losses during crop development. For phenotyping approaches to be approved by both breeders and producers, cost-effectiveness is a very important criterion for adopting the technique. Sustainable intensification in cultivation allows bred cultivars to obtain an optimum yield capacity, ensure more accurate crop management, and reduce the use of fertilizers, chemicals, and irrigation [112]. Phenotyping also focuses on optimizing predictable yields and maintaining the efficiency of crops [3]. In comparison to plant breeding, precise agriculture is used to enhance management methods, which results in the collection of suitable genotypes [113]. The main challenges that can be addressed by field phenotyping are irrigation, fertilization, and disease control to secure crop production. In phenotyping, various monitoring techniques for the evaluation, severity, and recording of plant health are essential. Research studies on the development of a suitable combination of sensors, vehicles, and analysis methods to customize phenotyping, based on specific interests, are promising in this sector [4].

In plants, phenotypic variation can be the result of a complex web of interactions between genotypes and the environment [114]. Therefore, precision phenotyping can determine which component in the set of phenotypes can predict specific parameters in the field, and the acquisition of phenotypic data allows a more precise description of G × E interactions [7]. Precise phenotyping facilitates the identification of quantitative trait loci, which can govern yield across different environmental conditions; but within the context of a more effective transition from the results of stress-related studies to improved cultivars, the difficulty of implementing accurate phenotyping is a widely recognized obstacle [115]. Thus, phenotyping techniques capable of revealing plant responses to environmental challenges, in both in-lab and field experimental conditions, are required [116]. In a more general sense, as they represent the technical basis of plant phenomics, precision phenotyping methods are both intensively and extensively required.

## 4. Image Processing Software

Manually monitoring crop growth stages and vegetation features on a broad scale is a time-consuming and laborious technique, and it is difficult to sample the entire area. Nowadays, unmanned aerial vehicles (UAVs) are being used more than satellites and manned aircraft, because these have a lower spatial and temporal resolution and a higher equipment cost [117]. As a result, UAVs have gained significant popularity, and their use for different applications related to agriculture, to obtain the required information about crops and field status is widely studied [118,119,120,121,122]. Furthermore, imaging sensors characterized by a low weight and high spatial and temporal resolutions have been developed for UAVs in recent years [123,124], and UAV imaging techniques are readily available, due to the upgrading of imaging and computation programs [125]. Table 2 presents recent photogrammetry and mapping software that have been integrated into UAVs, and describes their key parameters, this information was compiled based on the scientific papers published from 2007 to 2020. These techniques, which were previously only limited to other aerial vehicles, are now being used on UAVs, resulting in the promotion of frequent surveying of agricultural fields at a reasonable cost [125].

The expanding capabilities of UAVs, enhanced by the use of geographic information systems (GIS) and by technological breakthroughs in imaging methods, have increased the importance of this type of vehicle in precision agriculture [126]. Table 3 lists some of the recent pieces of GIS software that have been integrated into UAVs and describes their key features based on the prime scientific publications related to UAVs and remote sensing from 2007 to 2020. The change in vegetation features over time and space can be detected with an UAV fitted with a multispectral camera [126]. In healthy plants, a high reflection of infrared light and low reflection of red light were observed, whereas the opposite pattern (i.e., high reflection of red, and low reflection of infrared light) was revealed in unhealthy plants [126]. UAVs equipped with various sensors assist farmers in monitoring the health status of crops, irrigation requirements, insect pest incidence, and time of harvesting, and enable them to instantly take steps toward such issues [127,128]. HTP using a commercial UAV platform requires the selection of phenotypic traits for the breeding of target plants and, based on this, the appropriate sensor and UAV are then selected [118,119,120]. Nuijten et al. (2019) used a drone with high-resolution optical imagery to comprehend the potential of drone data for the evaluation of crop productivity on a large scale.

Remote sensing technology has become one of the most promising HTP technologies, as it provides nondestructive measurements of crop performance in both controlled and field environments [16]. In addition to its high flexibility and easy operation, remote sensing also helps to extract crop phenotypic information quickly and precisely [129]. Thus, remote sensing imaging products with a high spatial and temporal resolution and a light weight, have proven to be a valid technology and a suitable aerial platform for precision agriculture and plant breeding [16].

Finally, novel advances in remote sensing technology are expected to be quickly accepted and implemented by UAV platforms, making them one of the most important applications in HTP in the near future.

**Table 3 plants-11-01712-t003:** List of recent geographic information system software used for unmanned aerial vehicles (UAVs).

Geographic Information System Software	Features	Manufacturer
ArcGIS	Easy operation, compatible with the model builder or Python; supports visualization, analysis, and maintenance of data in 2D, 3D, and 4D; wide range of data sharing; users can operate the ArcGIS system via Web GIS	ESRI
QGIS	Features are similar to those of ArcGIS; supports both bitmap and vector layers; it is free and open-source	QGIS
ERDAS Imagine	Simplifies image processing; includes a wide range of tools for the analysis of image data; allows graphical editing; includes hyperspectral and multispectral data tools and LiDAR tools; allows spatial modeling	Hexagon geospatial [130]
ENVI	Easy to use; supports visualization, processing, and analysis of all types of geospatial data	L3Harris geospatial [131]
Grass GIS	License-free use, open-source; compatible with SQL programming language, geocoding, 2D, and 3D raster analysis; provides LiDAR tools, raster, and vector statistics	Grass Development Team [132]

## 5. Challenges in High Throughput Phenotyping

The major purpose of phenotyping techniques is to incorporate different phenotypic methods for evaluating agronomic traits, abiotic and biotic stress, and also assess factors that contribute to the yield potential of the crop in order to increase the genetic potential and use in crop improvement. In general, HTP in the plant is still challenging, as the development and utilization of techniques for the accurate recording of important agronomic, biotic and abiotic traits, and monitoring of crops have not been promoted to their full potential. In this context, there are several challenges for the HTP, for instance, complex/quantitative traits, root phenotyping, environmental effect, multilocation and replicated trials field plot measurement. Particularly, employing the HTP at the spatial and temporal resolutions of tissue or cellular level, morphologies, micro phenotypes, and below the soil traits is very challenging [133]. The major challenges faced by HTP are the identification of nondestructive, easy, operational, highly repeatable, robust, efficient, low-cost, and fast phenotypic tools. Moreover, a huge quantity of data is generated by HTP so storing, managing, processing these HTP data and making meaningful biological information are also very challenging [94,134]. However, with the emerging new tools, researchers have proposed the utilization of VIs, portable tools, multisensor-based hyperspectral imaging, and their utilization in plant HTP to counter the mentioned challenges. UAVs have a great potential to determine plants’ phenotypic trait differences among crops and, at the same time, they allow the collection of different information about vegetation across wide regions rapidly and cost-effectively, without destroying plants. Vis integrated with UAVs are attracting increasing attention, and researchers are more interested in implementing these technologies for the HTP of diverse plant species. Similarly, several publicly available open-source online databases and repositories (AraPheno, Cleared Leaves DB, PHENOPSIS DB, PhenoFront, and plant genomics and phenomics research data repository) were developed to store, manage, and access HTP big data [133]. Besides, novel imagery-based two-dimensional (2D) and three-dimensional (3D) tools (RootSlice and RootScan) along with X-ray, microcomputed tomography and computational software (RootReader2D, RhizoTubes, WinRHIZO, SmartRoot, and RootAnalyzer) have been introduced to analyze plant root traits [135,136,137,138]. Recent reviews [135,136,137,139] have described the different HTP platforms and the need for a multidomain approach to address the challenges faced by HTP. With the advancement of HTP, there seems to be no doubt that plant phenomics has many challenges, and these need to be addressed in the coming future.

## 6. Future Prospects

In this study, we reviewed various VIs and UAV platforms used for HTP in agricultural research. Plant phenotyping with the advancement of high throughput is on the verge of entering the big-data era. Individual phenotypic information is not sufficient for association analyses. Therefore, complete phenomics information, multiscale (physiology, structure, omics, genomics, and environmental) interaction will be the foundation of research in the future. In light of the emerging environmental challenges, future research in plant phenotyping needs to develop new cost-effective technologies based on artificial intelligence/remote sensing for the advancement of image-based phenotyping. Deep or machine learning modeling, simulation tools are also needed to consider the development of new applications of HTP. In addition, it is also important to identify a proper automated phenotyping system to accurately work across plant species and draw reliable conclusions based on large sample statistics and the analysis of relationships with agronomic traits. In this context, the importance of collaboration and data sharing at the national and international levels cannot be overstated. Thus, research on the rapid advancement of genetic improvement in breeding programs, and on reliable, automatic, comprehensible, and multifunctional HTP techniques is required. We are already aiming to obtain some of these important techniques in our laboratory, to assist conventional breeding in the restructuring of existing phenotypic breeding processes and allow for a gradual shift to a more high-throughput, data-driven approach.

## Figures and Tables

**Figure 1 plants-11-01712-f001:**
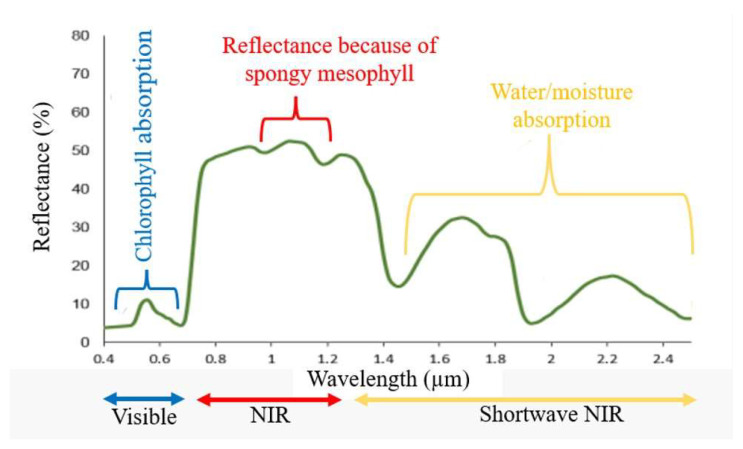
Different spectral reflectance curves for vegetation, modified from [15]. The main absorption and reflectance characteristics are represented.

**Table 1 plants-11-01712-t001:** Vegetation index types and their equation and utilization in plant phenotyping.

Index/Abbreviations	Formula	Utilization	Crops	Reference
Anthocyanin reflectance index 1 (ARI1)	$\mathrm{ARI}1=\frac{1}{R_{550}}-\frac{1}{R_{700}}$	Estimates anthocyanin accumulation in leaves	Maple, cotoneaster, dogwood, and pelargonium	[21]
Anthocyanin reflectance index 2 (ARI2)	$\mathrm{ARI}2=R_{800}\left[\frac{1}{R_{550}}-\frac{1}{R_{700}}\right]$	Estimates anthocyanin accumulation in leaves	Maple, cotoneaster, dogwood, and pelargonium	[21]
Atmospherically resistant vegetation index (ARVI)	$\mathrm{ARVI}=\frac{R_{800\;}+\left[R_{680\;}-\gamma \left(R_{450}-R_{680}\right)\right]}{R_{800}+\left[R_{680}-\gamma \left(R_{450}-R_{680}\right)\right]}$	Utilized for remote sensing of vegetation and atmospheric effect	—	[22]
Carotenoid index (CARI)	$\mathrm{CRI}2\;=\;\frac{R_{720}}{R_{510}}-1$	Detects leaf carotenoid content	Winter wheat	[23]
Carotenoid reflectance index 1 (CRI1)	$\mathrm{CRI}1\;=\;\frac{1}{R_{510}}-\frac{1}{R_{550}}$	Detects leaf carotenoid content	Norway maple, chestnut, and beech	[24]
Carotenoid reflectance index 2 (CRI2)	$\mathrm{CRI}2=\frac{1}{R_{510}}-\frac{1}{R_{700}}$	Determines carotenoid content in leaves	Norway maple, chestnut, and beech	[24]
Red-edge chlorophyll index	$\mathrm{CIrededge}=\frac{RNIR}{Rred\;edge}-1$	Determines chlorophyll content in both anthocyanin-containing and anthocyanin-free leaves	Soybean and maize	[25,26]
Carter stress index (CTR1)	$\mathrm{CTR}1=\frac{R_{695}}{R_{420}}$	Utilized to derive the chlorophyll content of winter wheat under stripe rust stress	Wheat	[27]
Carter stress index (CTR2)	$\mathrm{CTR}2=\frac{R_{695}}{R_{760}}$	Detects the nutritional status of crops	Maize	[28]
Cellulose absorption (CAI)	$\mathrm{CAI}=0.5\;\left(R_{2000}+R_{2200}\right)-R_{2100}$	Estimates crop residue cover	Corn, soybean, and wheat	[29,30]
Dual-polarization SAR vegetation index (DPSVI)	$\mathrm{DPSVI}=\frac{{}_{\sigma_{vh\left(i\right)}\left[\begin{array}{c} \;\left(\sigma_{vv\left(max\right)}\sigma_{vh\left(i\right)}-\sigma_{vv\left(i\right)}\sigma_{vh\left(i\right)}+\sigma^{2}{}_{vh\left(i\right)}\right)+\\ \left(\sigma_{vv\left(max\right)}\sigma_{vv\left(i\right)}-\sigma^{2}{}_{vv\left(i\right)}+\sigma_{vh\left(i\right)}+\sigma_{vv\left(i\right)}{}_{}\right)\; \end{array}\right]}}{\sqrt{2}\times \;\sigma_{vv\left(i\right)}}$	Utilized to estimate biomass and vegetation	Elephant foot yam, turmeric, onion, grass, cassava manioc, millet crops, and black gram	[31,32]
Enhanced vegetation index (EVI)	EVI=2.5× $\;\frac{\left(NIR-RED\right)}{\left(NIR+6\;\;\times \;RED-7.5\;\;\times \;Blue+1\right)}$	Utilized to assess canopy structural variations, leaf area, canopy type, plant physiognomy, and canopy architecture	Grass/shrub, savanna, and tropical forest biomes	[14]
Growing degree days mid-stay-green index (GDD_midstg,i_)	${\mathrm{GDD}}_{\mathrm{midstg,i}}\;=\;{\mathrm{GDD}}_{\mathrm{Heading,i}}\;+\;\frac{{\mathrm{OnSen}}_{\mathrm{GDDAS,i}}-{\mathrm{GDD}}_{\mathrm{Heading,i}}}{2}$	Utilized for biomass determination, canopy temperature, and greenness indicator	Wheat	[33]
Gitelson and Merzlyak index (GM1)	$\mathrm{GM}1=\frac{R_{750}}{R_{550}}$	Measures biophysical parameters	Sunflower	[34]
Gitelson and Merzlyak index (GM2)	$\mathrm{GM}2=\frac{R_{750}}{R_{700}}$	Detects drought stress	Wheat	[35]
Global environmental monitoring index (GEMI)	$\mathrm{GEMI}=\mathrm{eta}(1\;-\;0.25\times \mathrm{eta})\;-\;\frac{\left(RED-0.125\right)}{\left(1-Red\right)}$ $\mathrm{where}\;\mathrm{eta}\;=\;\frac{2\left(NIR^{2}-RED^{2}\right)+1.5\;\times \;NIR+0.5\;\times \;RED}{NIR+RED+0.5}$	Useful to compare observations under varying atmospheric and illumination conditions; more sensitive to actual surface conditions than SR or NDVI over the bulk of the range of vegetation conditions	—	[36]
Green atmospherically resistant index (GARI)	$\mathrm{GARI}=\frac{NIR-\left[Green-\gamma \left(Blue-Red\right)\right]}{NIR+\left[Green-\gamma \left(Blue-Red\right)\right]}$	Measures the rate of photosynthesis and monitors plant stress	Maple, chestnut	[37]
Green chlorophyll index (GCI)	$\mathrm{GCI}=\left(\frac{RNIR}{RGreen}\right)\times -1$	Measures leaf chlorophyll and carotenoids content	Maple, chestnut, wild vine, and beech	[38]
Green difference vegetation index (GDVI)	GDVI=NIR−Green	Utilized to improve in-season estimates of N requirements	Corn	[39]
Green leaf index (GLI)	$\mathrm{GLI}=\frac{\left(Green-Red\right)+\left(Green-Blue\right)}{\left(2\times \;Green\right)+Red+Blue}$	Utilized to map and document the extent and intensity of goose impacts on wheat fields	Wheat	[40]
Green optimized soil-adjusted vegetation index (GOSAVI)	$\mathrm{GOSAVI}=\frac{NIR-Green}{NIR+Green+0.16}$	Utilized to improve in-season estimates of N requirements	Corn	[39]
Green ratio vegetation index (GRVI)	$\mathrm{GRVI}=\frac{NIR}{Green}$	Utilized to improve in-season estimates of N requirements	Corn	[39]
Green soil-adjusted vegetation index (GSAVI)	$\mathrm{GSAVI}=1.5\;\times \;\frac{\left(NIR-Green\right)}{\left(NIR+Green+0.5\right)}$	Utilized to improve and predict nitrogen requirements	Corn	[39]
Green vegetation index (GVI)	$\mathrm{GVI}=\left(-0.2848\;\times \;TM_{1}\right)+\;\left(-0.2435\;\times \;TM_{2}\right)+\;\left(-0.5436\;\times \;TM_{3}\right)+\;\left(0.7243\;\times \;TTM_{4}\right)+\;(0.0840\;\times \;TM_{5})+\;\left(-0.1800\;\times \;TM_{7}\right)$	Minimizes the effects of background soil while emphasizing green vegetation to estimate and correct atmospheric haze and moisture effects	Wheat	[41]
Greenness index (G)	$G=\frac{R_{554}}{R_{667}}$	Evaluates corn nitrogen status under different sulfur levels	Corn	[42]
Hyperspectral narrow bands (HNB)	$\mathrm{HNB}=\frac{R2-R1}{R1+R2}$	Utilized to determine biophysical (biomass, leaf area index) and biochemical quantities (leaf nitrogen and plant pigments)	Wheat, maize, rice, barley, soybeans, pulses, cotton, and alfalfa	[43,44]
Hyperspectral vegetation indices (HVI*ij*)	$\mathrm{HVIij}=\frac{Rj-Ri}{Rj+Ri}$	Utilized to determine biophysical (biomass, leaf area index) and biochemical quantities (leaf nitrogen and plant pigments)	Wheat, maize, rice, barley, soybeans, pulses, cotton, and alfalfa	[43,44]
Infrared percentage vegetation index (IPVI)	$\mathrm{IPVI}=\frac{NIR}{NIR-Red}$	Significant in the monitoring of global biomass	—	[45]
Index R_780/_R_700_ and R_780/_R_740_	$I=\frac{R_{780}\;}{R_{700}\;}$ $\;\mathrm{and}\;\frac{R_{780}\;}{R_{740}\;}$	Determine the above nitrogen and biomass	Wheat	[46]
Leaf area index (LAI)	LAI=(3.618 ×EVI−0.118)	Utilized to estimate foliage cover and to forecast crop growth and yield	Winter barley and wheat, spring barley, peas, grass, maize, and beets	[47]
Lichtenthaler index (LIC1)	$\mathrm{LIC}1=\frac{R_{790}-R_{680}}{R_{790}+R_{680}}$	Detects bacterial wilt disease in Brinjal	Brinjal	[48]
Lichtenthaler index (LIC2)	$\mathrm{LIC}2=\frac{R_{440}}{R_{690}}$	Measures leaf nitrogen content	Wheat	[49]
Lignin cellulose absorption index (LCAI)	$\mathrm{LCAI}=100\;\times \;\left[\left(R_{2185\;to\;2225}-R_{2145\;to\;2185}\right)+\left(R_{2185\;to\;2225}-R_{2295\;to\;2365}\right)\right]$	Utilized to quantify crop residue cover and classify tillage intensity over diverse lands	Corn, soybean, wheat, tall fescue, and alfalfa	[50]
Modified chlorophyll absorption in reflectance index (MCARI)	$\mathrm{MCARI}=[\left(R_{700}-R_{670}\right)-0.2$ × $\;\left(R_{700}-R_{550}\right)]$ × $\;\left(\frac{R_{700}}{R_{670}}\right)$	Utilized to identify in-field heterogeneity for field segmentation as a feature of plant morphology and chlorophyll stress status	Cotton	[51]
Modified chlorophyll absorption in reflectance index (MCARI1)	MCARI1=1.2$\;\times \;\;[2.5\;\times \;\left(R_{790}-R_{670}\right)$−$1.3\;\times \;\;(R_{790}-R_{550}$)]	Determines nitrogen and chlorophyll levels in bean-plant leaves	Bean	[52]
Modified chlorophyll absorption ratio index improved (MCARI2)	$\mathrm{MCARI}2=\frac{1.5\;\left[2.5\;\left(R_{800}-R_{670}\right)-1.3\;\left(R_{800}-R_{550}\right)\right]}{\sqrt{{\left(2\;\times R_{800}+1\right)}^{2}\;}-\left(6\;\times R_{800}-\sqrt{R_{670}}\right)-0.5}$	Predicts the LAI of crop canopies	Corn, wheat, and soybean	[53]
Modified nonlinear index (MNLI)	$\mathrm{MNLI}=\frac{\left(NIR^{2}-Red\right)\times \left(1+L\right)}{NIR^{2}+Red+L}$	Utilized to interpret soil background and improve crop discrimination, crop yield, crop stress, pest/disease surveillance, and disaster management	—	[54]
Modified red-edge normalized difference vegetation index (MRENDVI)	$\mathrm{MRENDVI}=\frac{R_{750}-R_{705}}{\rho_{750}+\rho_{705\;}-2\times \;\rho_{445}}$	Utilized to estimate chlorophyll content and determine the variation in reflectance caused by chlorophyll absorption	Eucalyptus	[55]
Modified red-edge simple ratio (MRESR)	$\mathrm{MRESR}=\frac{R_{750}-R_{445}}{R_{705}-R_{445}}$	Utilized to estimate pigment content and carotenoid/chlorophyll ratios in green leaves	Wide species	[56]
Modified simple ratio (MSR)	$\mathrm{MSR}=\frac{\left(\frac{NIR}{Red}\right)-1}{\left(\sqrt{\frac{NIR}{Red}}\right)+1}$	Advantageous in field data evaluation and less sensitivity to canopy optical and geometrical properties	Jackpine and black spruce	[57]
Modified soil-adjusted vegetation index 2 (MSAVI2)	$\mathrm{MSAVI}2=\frac{2\times NIR+1-\sqrt{{\left(2\;\times NIR+1\right)}^{2}-8\left(NIR-Red\right)}}{2}$	Reduces soil noise and increases the dynamic range of the vegetation signal	Cotton	[58]
Modified triangular vegetation index improved (MTVI2)	$\mathrm{MTVI}2=\frac{1.5\;\left[\;1.2\left(R_{800}-R_{550}\right)-2.5\;\left(R_{670}-R_{550}\right)\right]}{\sqrt{{\left(2\;\times \;R_{800\;}+1\right)}^{2}}-\left(6\;\times \;R_{800}-5\times \;\sqrt{R_{670}}\;\right)-0.5}$	Determines chlorophyll content variations and linearly related to green LAI	Corn, wheat, and soybean	[53,59]
Modified triangular vegetation index (MTVI)	$\mathrm{MTVI}=1.2[1.2\left(R_{800}-R_{550}\right)-2.5\left(R_{670}-R_{550}\right)$	Determines chlorophyll content variations and linearly related to green LAI	Corn, wheat, soybean	[53]
Moisture stress index (MSI)	$\mathrm{MSI}=\frac{R_{1599}}{R_{819}}$	Determines leaf and canopy water content	California live oak, blue spruce, sweetgum, soybean, maple, apricot, mulberry, and cherry laurel	[60,61]
Nonlinear index (NLI)	$\mathrm{NLI}=\frac{NIR^{2}-Red}{NIR^{2}+Red}$	Linearizes relationships with surface parameters that tend to be nonlinear	Corn and aspen	[62]
Normalized difference infrared index (NDII)	$\mathrm{NDII}=\frac{R_{819}-R_{1649}}{R_{819\;}+R_{1649}}$	Estimates leaf and canopy water content, and correlates spectral response to ambient soil salinity	Smooth cordgrass	[63]
Normalized difference lignin index (NDLI)	$\mathrm{NDLI}=\frac{\log(\frac{1}{R_{1754}})-\log\left(\frac{1}{R_{1680}}\right)}{\log(\frac{1}{R_{1754}})+\log\left(\frac{1}{R_{1680}}\right)}$	Determines biochemical concentration (nitrogen and lignin) and canopy structural features	Wide species	[64,65,66]
Normalized difference nitrogen index (NDNI)	$\mathrm{NDNI}=\frac{\log(\frac{1}{R_{1510}})-\log\left(\frac{1}{R_{1680}}\right)}{\log(\frac{1}{R_{1510}})+\log\left(\frac{1}{R_{1680}}\right)}$	Determines biochemical concentration (nitrogen and lignin) and canopy structural features	Wide species	[64]
Normalized difference vegetation index (NDVI)	$\mathrm{NDVI}=\frac{R_{NIR}-R_{RED}}{R_{NIR}+R_{RED}}$	Predicts chlorophyll content in rice plants under stress from heavy metal condition	Rice	[67]
Normalized difference water index (NDWI)	$\mathrm{NDWI}=\frac{\left(R_{857}-R_{1241}\right)}{\left(R_{857\;}+{\;}_{R1241}\right)}$	Detects vegetation liquid or water content	Corn, soybean, and redwood	[68,69]
Normalized multiband drought index (NMDI)	$\mathrm{NMDI}=\frac{R_{860}-(R_{1640}-R_{2130})}{R_{860}+(R_{1640\;}+R_{2130})}$	Monitors soil and vegetation moisture from space, detects fires	—	[70,71]
Normalized phaeophytization index (NPQI)	$\mathrm{NPQI}=\frac{R_{415}-R_{435}}{R_{415}+R_{435}}$	Detects mite effects on apple trees	Apple	[72]
Normalized pigment chlorophyll index (NPCI)	$\mathrm{NPCI}=\frac{R_{680}-R_{430}}{R_{680}+R_{430}}$	Evaluates chlorophyll loss and leaf senescence caused by aphid feeding	Wheat	[73]
Plant biochemical index (PBI)	$\mathrm{PBI}=\frac{R_{810}}{R_{560}}$	Determine total chlorophyll and nitrogen concentrations	Rice, sorghum, mung bean, and pigeon pea	[74]
Photochemical reflectance index (PRI)	$\mathrm{PRI}=\frac{R_{531}-R_{570}}{R_{531}+R_{570}}$	Rapidly evaluates leaf water status to estimate the water stress index of crops	Quinoa	[72]
Plant senescence reflectance index (PSRI)	$\mathrm{PSRI}=\frac{R_{680}-R_{500}}{R_{750}}$	Estimates pigment content depending on the onset, stage, relative rates, and kinetics of leaves and fruits	Maple, chestnut, potato, coleus, lemon, and apple	[75]
Reflectance at 1200 nm (Ratio_1200)_	${\mathrm{Ratio}}_{1200}\;=\;\frac{2\;\times \;Avg\;(R_{1180\;to\;1220)}}{Avg\;(R_{1090\;to\;1110)+Avg\;(R_{1265\;to\;1285)}}}$	Determine shoot biomass, phenology, morphology, and canopy structural parameters	Wheat	[33]
Red-edge normalized difference vegetation index (RENDVI)	$\mathrm{RENDVI}=\frac{R_{750}-R_{705}}{R_{750}+R_{705}}$	Quantitative estimation of pigments	Horse chestnut and Norway maple	[76]
Red-edge position index (REPI)	—	Estimates chlorophyll concentration in fields	Slash pine	[77]
Red-green ratio index (RGRI)	$\mathrm{RGRI}=\frac{\sum_{i=600}^{699}R_{i}}{\sum_{j=500}^{599}R_{j}}$ or *R_RED_:R_GREEN_*	Determines chlorophyll, xanthophyll cycle, anthocyanin contents, and the change in photochemical	Sunflower, Douglas, and coast live oak	[78]
Renormalized difference vegetation index (RDVI)	$\mathrm{RDVI}=\frac{\left(NIR-Red\right)}{\sqrt{\left(NIR+Red\right)}}$	Utilized to investigate healthy vegetation, suitable for larger vegetation coverages and denser canopies	Corn, wheat, and soybean	[53,79]
Simple ratio index (SR)	$\mathrm{SR}=\frac{R_{NIR}}{R_{RED}}$	Utilized to estimate crop growth and to forecast grain yield	Wheat	[80]
Simple ratio pigment index (SRPI)	$\mathrm{SRPI}=\frac{R_{430}}{R_{680}}$	Utilized for large-area monitoring of plants’ N status in wheat, allows the precise application of fertilizers	Wheat	[81]
Soil-adjusted vegetation index (SAVI)	$\mathrm{SAVI}=\frac{1.5\;\times \left(NIR-Red\right)}{\left(NIR+Red+0.5\right)}$	Best used in areas with relatively sparse vegetation where the soil is visible through the canopy	Cotton and range grass	[82]
Structure-insensitive pigment index (SIPI)	$\mathrm{SIPI}=\frac{R_{800}-R_{445}}{R_{800}-R_{680}}$	Detects pest damage on wheat through the identification of chlorophyll loss and estimates carotenoids and chlorophyll content ratio	Maize, wheat, tomato, soybean, sunflower, and sugar beet	[83,84]
Sum green index (SGI)	−	Detects changes in vegetation greenness and vegetation canopy opening	Wheat	[85]
Standardized LAI determining index (sLAIDI)	$\mathrm{sLAIDI}=\left(\frac{R_{1050-R1250}}{R_{1050+R1250}}\right)$	Determine LAI, more accurately extract biochemical parameters	Apple, peach, citrus, and orchard	[86,87]
Transformed chlorophyll absorption reflectance index (TCARI)	$\mathrm{TCARI}=3\left[\left(R_{700}-R_{670}\right)-0.2\left(R_{700}-R_{550}\right)\left(\frac{R_{700}}{R_{670}}\right)\right]$	Predicts the LAI of crop canopies	—	[53]
Triangular vegetation index (TVI)	$\mathrm{TVI}\;\frac{R120\left(R_{750}-R_{550}\right)-200\left(R_{670}-R_{550}\right)}{2}$	Estimates the green leaf area index and canopy chlorophyll density	—	[88]
Transformed difference vegetation index (TDVI)	$\mathrm{TDVI}=1.5\left[\frac{\left(NIR-Red\right)}{\sqrt{NIR^{2}}+Red+0.5}\right]$	Useful for monitoring vegetation cover in urban environments	Cotton	[89]
Triangular greenness index (TGI)	$\mathrm{TGI}=\frac{\left(\;\lambda_{Red}-\lambda_{Blue\;}\right)\left(R_{Red}-R_{Green}\right)-\left(\;\lambda_{Red}-\lambda_{Green}\right)\left(\rho_{Red}-\rho_{Blue}\right)}{2}$	Detects leaf chlorophyll content and green vegetation	Corn, soybean, sorghum, dandelion, sweetgum, tulip tree, and small-leaf linden	[90]
Visible atmospherically resistant index (VARI)	$\mathrm{VARI}=\frac{Green-Red}{Green+Red-Blue}$	Estimates the fraction of vegetation in environments with low sensitivity to atmospheric effects	Wheat	[24]
Vogelmann red-edge index 1 (VREI1)	$\mathrm{VREI}1=\frac{R_{740}}{R_{720}}$	Quantifies leaf-level chlorophyll content	Sugar maple	[91]
Vogelmann red-edge index 2 (VREI2)	$\mathrm{VREI}2=\frac{R_{734}-R_{747}}{R_{715}+R_{726}}$	Quantifies leaf-level chlorophyll content	Sugar maple	[91]
Water band index (WBI)	$\mathrm{WBI}=\frac{R_{970}}{R_{900}}$	Estimates water status and relative water content	Gerbera, pepper, and bean	[92,93]
Wide dynamic range vegetation index (WDRVI)	$\mathrm{WDRVI}=\frac{\left(a\;\times NIR-Red\right)}{\left(a\;\times \;NIR+Red\right)}$	Aids in the robust characterization of crops physiological and phenological characteristics, dry matter content, water content, leaf mesophyll structure index, and some other less influential factors	Wheat and maize	[24,94]
World view improved vegetation index (WV-VI)	$\mathrm{WV}-\mathrm{VI}=\frac{\left(NIR^{2}-Red\right)}{\left(NIR^{2\;}+Red\right)}$	Determines moisture content, vegetation health; distinguishes natural features from man-made objects and supports land mapping	—	[95]
Zarco-Tejada and Miller (ZMI)	$\mathrm{ZMI}=\frac{R_{750}}{R_{710}}$	Estimates leaf nitrogen	Potato	[96]

Note: ‘—’ Information is not available, all the equation details can be found in the respective publication.

**Table 2 plants-11-01712-t002:** List of photogrammetry and mapping software used for unmanned aerial vehicles (UAVs).

Photogrammetry and Mapping Software	Features	Manufacturer
Agi Soft photoscan Pro	Simple interface, easy to learn for beginners; supports Python script; distributed, flexible, nonlinear processing; elaborate model editing for accurate results; includes inbuilt tools to measure distances, areas, and volumes; produces better quality point clouds, digital elevation models, orthoimage generation, export, makes accurate measurements; access to 3D modeling presents a wider range of panorama stitching; processes RGB, NIR, thermal, and multispectral imagery	Agisoft
Precision Mapper	A better option for the agricultural sector; includes tools to analyze crop health, obtain volumetric measurements, generate orthoimages, point clouds, and 3D models; offers NDVI enhancements, a canopy cover calculator, and even finds standing water in a field	Precision Hawk
Maps Made Easy	Free package, easy to operate; generates 3D models, accurate calculations, and NDVI maps; allows the user to collage, classify, and add a comment to NDVI maps easily	DRONES MADE EASY
Drone Deploy Field scanner	Easy to operate, offers cloud-based processing; supports DJI drones; allows DTM (digital terrain models), 3D models, and orthomosaic generation; collects NDVI data, and allows the quick view of NDVI maps even without internet connection	Drone Deploy
PiX4d	Easy to use; supports cameras with a wider range; provides automated computing such as orthoimage, DSM generation, and cloud service; image data can be processed even without internet connection	PiX4d
Sentera Ag Vault	Features are similar to those of Pix4D and Drone Deploy; allows data collection for NIR, NDVI, and NDRE; allows the instant view of RGB and NIR data after capture; quick tile images can be generated in the field even without internet connection	SENTERA
Botlink Mapper	Creates high-definition maps, VI maps, and terrain maps to help find wet and dry areas; NIR photography helps to monitor and analyze the crops’ health status; allows easy collage of aerial images into a single, high-definition map; creates stunning digital surface and 3D models to identify high and low points or drainage issues	Botlink
Icaros OneButton	Easy to use, very high-quality processing, and program to monitor UAV flight period	Icaros

## Data Availability

Not applicable.

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
