# Peer review of "Utilization of Spectral Indices for High-Throughput Phenotyping"

_plants, 2022, doi:10.3390/plants11131712_

Round 1
Reviewer 1 Report
Overall, this is a clear, concise and well-written manuscript. The introduction is pertinent and based on interesting papers.
The procedure is described in details and gives sufficient information on the study logic.
In addition, the results are clear.
Kind Regards
Author Response
Reviewer 1
Comments and Suggestions for Authors
Overall, this is a clear, concise and well-written manuscript. The introduction is pertinent and based on interesting papers.
The procedure is described in details and gives sufficient information on the study logic.
In addition, the results are clear.
Kind Regards
Answer: We would like to thank the worthy reviewer for the time devoted to this manuscript. We highly appreciate your efforts and grateful for comments. All the changes that were made in the revised MS can be found with the track change. We have made changes in the numbering and corrected sections captions. All the changes that were made in the revised MS can be found with the highlight in green color.

Reviewer 2 Report
It is an excellent piece of review. The only problem that I see is that the authors have not made a correct structuring/numbering of the sections. The paper should be revised in such way that all sections should be numbered properly.
Please follow all the appropriate format for sections captions, figures, tables and references both in the text and the reference list.
Author Response
Comments and Suggestions for Authors
It is an excellent piece of review. The only problem that I see is that the authors have not made a correct structuring/numbering of the sections. The paper should be revised in such way that all sections should be numbered properly.
Please follow all the appropriate format for sections captions, figures, tables and references both in the text and the reference list.
Answer: We would like to thank the worthy reviewer for the time devoted to this manuscript. All the comments and suggestions were genuine. We highly appreciate your efforts and agree to the suggested changes. We have changed the numbering and corrected sections captions. All the changes that were made in the revised MS can be found with the highlight in green color.
We have made required changes to our manuscript, thanks to the editorial team and the reviewers for their valuable input and time devoted to the manuscript towards improvement. We hope that with this revision now this manuscript is in a good shape to be published in MDPI Plants.
Kind Regards,
Round 2
Reviewer 2 Report
The final version is an excellent contribution review paper on the use of spectral indices for high-throughput phenotyping. I don't have other comments. The paper can be published.